# High-Likelihood Area Matters — Rewarding Correct, Rare Class Predictions Under Imbalanced Distributions

## Abstract

Learning from natural datasets poses significant challenges for traditional classification methods based on the cross-entropy objective due to imbalanced class distributions. It is intuitive to assume that the examples from rare classes are harder to learn so that the classifier is uncertain of the prediction, which establishes the low-likelihood area. Based on this, existing approaches drive the classifier actively to correctly predict those incorrect, rare examples. However, this assumption is one-sided and could be misleading. We find in practice that the high-likelihood area contains correct predictions for rare class examples and it plays a vital role in learning imbalanced class distributions. In light of this finding, we propose the Eureka Loss, which rewards the classifier when examples belong to rare classes in the high-likelihood area are correctly predicted. Experiments on the large-scale long-tailed iNaturalist 2018 classification dataset and the ImageNet-LT benchmark both validate the proposed approach. We further analyze the influence of the Eureka Loss in detail on diverse data distributions.

## 1 Introduction

Existing classification methods usually struggle in real-world applications, where the class distributions are inherently imbalanced and long-tailed (Van Horn & Perona, 2017; Buda et al., 2018; Liu et al., 2019; Gupta et al., 2019), in which a few head classes occupy a large probability mass while most tail (or rare) classes only possess a few examples. The language generation task is a vivid example of the long-tailed classification. In this case, word types are considered as the classes and the model predicts probabilities over the vocabulary. Common words such as *the*, *of*, and *and* are the head classes, while tailed classes are rare words like *Gobbledygook*, *Scrumptious*, and *Agastopia*. Conventional classifiers based on deep neural networks require a large number of training examples to generalize and have been found to under-perform on rare classes with a few training examples in downstream applications (Van Horn & Perona, 2017; Buda et al., 2018; Cao et al., 2019).

It is proposed that the traditional cross-entropy objective is unsuitable for learning imbalanced distributions since it treats *each instance* and *each class* equivalently (Lin et al., 2017; Tan et al., 2020). In contrast, the instances from tail classes should be paid more attention, indicated by two main approaches that have been recently investigated for class-imbalanced classification: the frequency-based methods and the likelihood-based methods. The former (Cui et al., 2019; Cao et al., 2019) directly adjust the weights of the instances in terms of their class frequencies, so that the instances from the tail classes are learned with a higher priority no matter whether they are correctly predicted or not. The latter (Lin et al., 2017; Zhu et al., 2018) instead penalize the inaccurate predictions more heavily, assuming that the well-classified instances, i.e., the instances in the high-likelihood area, factor inconsequentially in learning imbalanced distributions.

However, neither of these two approaches realistically depicts the likelihood landscape. In particular, the high-likelihood area, where the classifier makes the correct predictions for both common class examples and rare class ones, contributes significantly to generalization. However, this area is not well-shaped, as illustrated in Figure 1. Specifically, the frequency-based methods imply an impaired learning of common class examples that are the principle part of the natural data, while the likelihood-

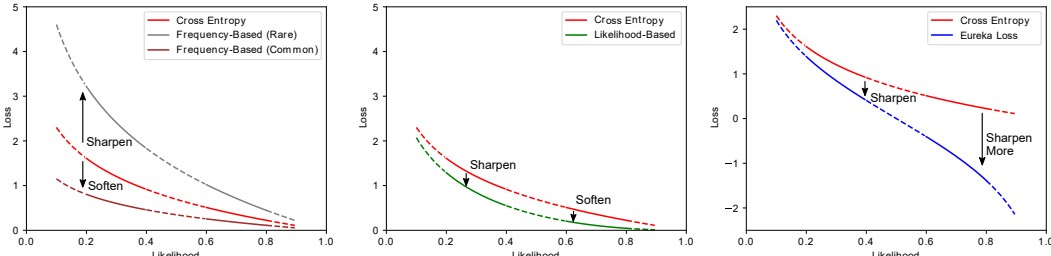

Figure 1: Conceptual illustration of approaches to learning unbalanced class distributions. For an instance in the training data, the frequency-based methods either sharpen or soften the loss for all likelihoods according to its class frequency, while the likelihood-based methods adjust the loss in the low- or high-likelihood area, respectively. The high-likelihood area is relatively deprioritized in both cases. The proposed Eureka Loss progressively rewards the systems with higher bonus for higher-likelihood.

based methods ignore the correctly-predicted rare class examples that can provide crucial insights into the underlying mechanism for predicting such examples.

In this paper, we first demonstrate that existing practice of neglecting predictions in the high-likelihood area is harmful to learning imbalanced class distributions. Furthermore, we find that simply mixing the cross-entropy loss and the Focal Loss (Lin et al., 2017) can induce substantially superior performance, which validates our motivation. In turn, we propose to elevate the importance of high-likelihood predictions even further and design a novel objective called Eureka Loss. It progressively rewards the classifiers according to both the likelihood and the class frequency of an example such that the system is encouraged to be more confident in the correct prediction of examples from rare classes. Experimental results on the image classification and the language generation tasks demonstrate that the Eureka Loss outperforms strong baselines in learning imbalanced class distributions.

Our contributions are twofold:

- We challenge the common belief that learning for examples in low-likelihood area is more important for learning tail classes and reveal that the correctly-predicted rare class examples make important contribution to learning long-tailed class distributions.

- We explore a new direction for learning imbalanced classification that focuses on rewarding correct predictions for tail classes examples, rather than penalizing incorrect ones. The proposed Eureka Loss rewards the classifier for its high-likelihood predictions progressively to the rarity of their class and achieves substantial improvements on various problems with long-tailed distributions.

## 2 RELATED WORK

**Frequency-based Data and Loss Re-balancing**    Previous literature on learning with long-tailed distribution mainly focusing on re-balancing the data distribution and re-weighting the loss function.

The former is based on a straightforward idea to manually create a pseudo-balanced data distribution to ease the learning problem, including up-sampling for rare class examples (Chawla et al., 2002), down-sampling for head class examples (Drummond & Holte, 2003) and a more concrete sampling strategy based on class frequency (Shen et al., 2016).

As for the latter, recent studies propose to assign different weights to different classes, and the weights can be calculated according to the class distribution. For example, Khan et al. (2018) design a cost-sensitive loss for major and minor class examples. An intuitive method is to down-weight the loss of frequent classes, while up-weight the contribution of rare class examples. However, frequency is not suitable to be directly treated as the the weight since there exists overlap among samples. An advancing alternative loss CB (Cui et al., 2019) proposes to calculate the effective number to substitute the frequency for loss re-weighting. However, since it assigns lower weight to head classes in the maximum likelihood training (Cross Entropy objective), it seriously impairs the learning

of head classes. Moreover, CB requires a delicate hyper-parameter tuning for every imbalanced distribution, leading to a lot of manul efforts. From the perspective of max-margin, a recent study LDAM (Cao et al., 2019) proposes to up-weight the loss of tail classes by a class-distribution based margin. Compared to the above methods, we choose to decrease the loss of tail classes by rewarding correct predictions rather than increasing the loss of tail classes through aggravated penalization.

**Deferring the Frequency-based Class-balanced Training**   Recent studies find that deferring the class-balanced training helps learn high-quality representations (Liu et al., 2019), and propose deferred Class-balanced training (deferred CB) (Cao et al., 2019), which chooses to adopt Cross Entropy objective at the beginning of training. Similarly, the Decoupling method (Kang et al., 2020) shows that the re-balancing strategies impair the quality of learned feature representations and demonstrate an improved performance learned with original data distribution, by training the model with Cross Entropy in the first phase and adopting class-balanced training in the second phase. This decoupling strategy can also be found in BBN (Zhou et al., 2019), which includes both class imbalanced and balanced training, and the transition from the former to the latter is achieved through a curriculum learning schedule. These methods achieve state-of-the-art performance in long-tailed classification. To be comparable with these methods and to analyse whether Eureka Loss is complementary to this technique, we propose deferred Eureka Loss, in which rewarding for rare class prediction is introduced to encourage the model to learn rare patterns when learning is stalled.

**Likelihood-based Loss**   Another dominant method for imbalanced classification is the likelihood-based method Focal Loss (FL)  (Lin et al., 2017), which proposes to down-weight the contribution of examples in the high-likelihood area. However, we argue that it is harmful for learning tail classes and choose an opposite direction by highlighting the high-likelihood area with a steeper loss.

**Transferring Representations**   Techniques for transferring information from sufficient head classes examples to under-represented rare classes examples belong to a parallel successful direction in this field. They include MBJ (Liu et al., 2020), which utilizes external semantic feature memory and FSA (Chu et al., 2020), which decomposes feature in to class-specific and class-generic components. These latest transfer learning based studies are less related to our paper but they also obtain good improvements in long-tailed classification, so we add them into comparison in the experiments.

# 3   ROLE OF THE HIGH-LIKELIHOOD AREA

The existing approaches to the long-tailed classification independently consider the class frequency and the example likelihood. However, we show that this one-sided reflection is problematic when dealing with the tail class examples that can be confidently classified. The tail class examples can be easily classified by the classifier, and the head class examples can also be hard for the classifier to recognize. The difficulty of classification depends on the inherent characteristic of the classes, rather than the sample size of the class. For example, in species classification, the *Portuguese man o'war* may be a rare class but can be easily classified due to its distinct features, compared to various kinds of *moths* which are common classes yet are hard to distinguish. However, the frequency-based methods continuously drive the classifier to fit the rare class examples, especially when they are difficult to predict, which may lead to overfitting. On the other hand, the likelihood-based methods relax the concentration in the high-likelihood area, which contains the tail class examples that are not hard to predict and provide insights to generalization.

To verify our point of view, we analyze the problem by dissecting the influence of the high-likelihood area with respect to the class frequency and demonstrate that properly encouraging the learning of well-classified tail class examples induce substantial improvements, before which we first give a brief introduction of classification with long-tailed class distributions.

## 3.1   PREPARATION: CLASSIFICATION WITH LONG-TAILED CLASS DISTRIBUTIONS

Let's consider the multi-class classification problem with the long-tailed class distribution. Given a class set $\mathbb{C}$, $n$ denotes the number of different classes in $\mathbb{C}$ and $m_i$ is the number of examples of the class $C_i$. For simplicity, we sort the class set $\mathbb{C}$ according to cardinal $m_i$ for $C_i$ such that $C_0$ is the class with the most examples and $C_{n-1}$ is the rarest class. Let $p$ be a $n$-dim probability vector predicted by a classifier model $f(x; \theta)$ based on the input $x$, where each element $p_i$ denotes the probability of the class $C_i$ and $y$ is a $n$-dim one-hot label vector with $y$ being the ground-truth class.

The probability vector can be calculated as

$$p = \sigma(f(x; \theta)),$$ (1)

where $\sigma$ is the normalizing function, e.g., softmax for multi-class classification. Typically, the parameters are estimated using maximum likelihood estimation (MLE), which is equivalent to using the Cross-Entropy Loss (CE) function, where the scalar $y \cdot \log p$ can be regarded as the (log)-likelihood:

$$\mathcal{L} = -\mathbb{E}_{(x,y) \in \mathcal{D}} \log p_{\text{model}}(y|x) = -\frac{1}{|\mathcal{D}|} \sum_{(x,y) \in \mathcal{D}} y \cdot \log p.$$ (2)

For deep neural network–based classifiers, due to the non-linearity of the loss function, the problem is typically solved by stochastic gradient descent, which requires the calculation of the gradient with respect to the parameters using the chain-rule, the process of which is called back-propagation:

$$\frac{\partial \mathcal{L}}{\partial \theta} = \frac{\partial \mathcal{L}}{\partial p} \frac{\partial p}{\partial \theta} = \frac{\partial \mathcal{L}}{\partial p} \frac{\partial \sigma(f(x; \theta))}{\partial \theta}.$$ (3)

We introduce the term **likelihood gradient** to denote $\partial \mathcal{L} / \partial p$, which modulates how the probability mass should be shifted and is a characteristic of the loss function instead of the classifier. For learning imbalanced class distributions, the common methods aim to shape the likelihood gradient so the rare classes are learned with priority, i.e., embodying a sharper loss and a larger likelihood gradient.

**Frequency-Based Methods**  Frequency-based methods alter the likelihood gradient according to the class frequencies, which are irrelevant to how well individual examples are classified. A simple form is using a $n$-dim weight vector $w$ composed of the class weights based on their frequencies in the dataset to determine the importance of examples from each class:

$$\mathcal{L} = -w_y \cdot (y \cdot \log p).$$ (4)

Note that when $w = 1$, it is identical to the cross-entropy objective. The weight vector is typically calculated as $w_i = \bar{m}/m_i$, where $\bar{m}$ is the average of $m_i$. As we can see, the standard weight is taken as the average of the class size, so that the classes with more examples are down-weighted and the classes with fewer examples are up-weighted. For a natural long-tailed distribution, the average is larger than the median, which suggests more classes are up-weighted. Advanced frequency-based methods try to obtain a more meaningful measurement of the class size, e.g., the Class-Balanced Loss (CB) proposed by Cui et al. (2019) utilizes an effective number $1-\beta^{m_i}/1-\beta$ for each class, where $\beta \in [0.9, 1)$ is a tunable class-balanced term.

**Likelihood-Based Methods**  Different from the frequency-based methods, likelihood-based methods adjust the likelihood gradient based on the instance-level difficulty as predicted by the classifier such that the examples in the low-likelihood area are more focused in training. For example, the well-known Focal Loss (FL) with a balanced factor $\alpha$ proposed by Lin et al. (2017) takes the following form:

$$\mathcal{L} = -\alpha(1 - p_y)^{\gamma_f} \cdot (y \cdot \log p),$$ (5)

where $\gamma_f > 0$, which controls the convexness of the loss and higher $\gamma_f$ indicates adjustment that are more significant. Note that when $\alpha = 1$ and $\gamma_f = 0$, it is identical to the cross-entropy objective. Following previous works (Cao et al., 2019; Liu et al., 2019; Cui et al., 2019), the $\alpha$ is set to 1 in multi-class classification, and Class-Balanced Focal Loss (FL+CB) can be viewed Focal Loss with uneven $alpha$ for each class in the multi-class setting. The key idea is to pay less attention to the well-classified examples and pay more attention to the badly-classified examples, because it is natural to assume the tail class examples are harder to learn and thus cannot be well-classified. However, such methods neglect the correctly-predicted tail class examples, the practice of which we show is not constructive to the learning of long-tailed class distributions.

## 3.2 Understanding the Influence of the High-Likelihood Area

To understand the influence of the high-likelihood area, we first prepare a variant of the Focal Loss, the Halted Focal Loss (HFL), such that the high-likelihood area is not deprioritized. The Halted Focal Loss reverts the Focal Loss to the Cross-Entropy Loss when the likelihood is high enough:

$$\mathcal{L} = \begin{cases} -\alpha(1 - p_y)^{\gamma_f} \cdot (y \cdot \log p), & \text{if } p_y \leq \varphi \\ -\alpha y \cdot [\log p + b], & \text{otherwise} \end{cases}$$ (6)

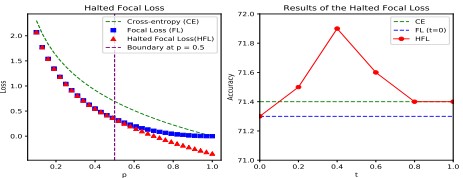

| Method | $AP$ | $AP_{50}$ | $AP_{75}$ |
|---|---|---|---|
| CE | 32.8 | 52.3 | 34.7 |
| FL | 33.8 | 52.5 | 35.9 |
| HFL | 34.0 | **52.7** | 36.2 |
| FL (Head) + HFL (Tail) | **34.2** | **52.7** | **36.3** |

Figure 2: Regaining focus on the high-likelihood area for the rare classes benefits classification. Left: Illustration of HFL which reverts FL to CE in the high-likelihood area. Right: Applying HFL only to the rare classes improves overall performance.

Table 1: Results of HFL on the COCO detection dataset, AP denotes average precision. As we can see, increasing the importance of the high-likelihood area achieves better results, and the main improvements come from the tail class examples.

where $p_y$ is prediction probability of the correct label, $b = \alpha(1 - (1 - \varphi)^{\gamma_f}) \log \varphi$ to ensure monotonicity and continuity, and $\varphi$ is the boundary between the low- and the high-likelihood area, which we set as $0.5$, i.e., a likelihood higher than which definitely renders a correct prediction. This mixed loss is plotted in left of the Figure 2, which has the same likelihood gradient as the cross-entropy in the high-likelihood area and remains the same as the Focal Loss in the low-likelihood area.

To decouple the effect of class frequency, we further explore to gradually transition from the Focal Loss to the Halted Focal Loss according to the class frequency of an example, e.g., from adopting the Halted Focal Loss only for the rarest class and the Focal Loss to other classes to adopting the Focal Loss only for the most common class and the Halted Focal Loss for the rest. Concretely, we set a proportion $t \in [0, 1]$ of classes to receive this loss and the remaining $1 - t$ proportion of classes adopt the original Focal Loss. The classes are ranked by inverse frequencies, such that the first class is the rarest class.

We conduct experiments on long-tailed CIFAR-10 using the aforementioned protocol to examine the effect of the high-likelihood area. The construction of the dataset is provided in Appendix C. We run each configuration 5 times with different random initialization and report the average test performance. The results are shown in the right of Figure 2.

As we can see, compared to the original Focal Loss, the proposed adaptation achieves better performance, indicating regaining focus in the high-likelihood area is beneficial. Nonetheless, the phenomenon can be also attributed to a better learning of the common classes instead of the rare classes. Our analysis based on the class frequency resolves this concern because the Halted Focal Loss brings more improvements if only tailed classes are learned this way, e.g., applying to the top-4 rare classes achieve the best overall performance, which proves that there are rare class examples that reside in the high-likelihood area and have non-negligible effect to generalization.

The importance of the high-likelihood area of the rare examples is further validated on the COCO detection dataset, where the classifier should determine whether the object appears in the image or not. The positive detection is the rare class since there are many false proposals. The experiment setting is in Appendix C. $AP_{50}$ and $AP_{75}$ measure the precision under different levels of overlap between predictions and ground-truth. As shown in Table 1, strengthening the high-likelihood area of the Focal Loss, especially for the rare class examples, obtains a more accurate and confident prediction.

## 4    EUREKA LOSS

We have shown that the high-likelihood area matters for long-tailed classification and in particular, the rare class examples in the area have pivotal contributions. Inspired by this finding, we propose to further enhance the importance of the high-likelihood area so that the likelihood gradient in the high-likelihood area can match or even surpass that in the low-likelihood area. Moreover, the adjustment is inline with the frequency of the class so the rarer the class, the larger the likelihood gradient. Extending the adjustment term $b$ in Eq. (6), we propose the **Eureka Loss** (EL):

$$\mathcal{L} = -\boldsymbol{y} \cdot \log \boldsymbol{p} - \text{bonus} \cdot \text{encouragement}, \tag{7}$$

where $-$bonus$\cdot$encouragement is intended to reward the well-classified rare examples. The bonus term depends on the example likelihood and the encouragement term depends on the class frequency. Different from the existing approaches that scale the Cross-Entropy Loss, punishing the incorrect predictions selectively, the proposed Eureka Loss deals with long-tailed classification from another perspective, rewarding the correct predictions progressively with their class frequencies.

**Bonus** indicates how well the system executes the task and is designed to be a function of the probability of the ground-truth class to reward the model when it makes correct prediction. In particular, in light of the findings discussed in Section 3.2, we propose to increase the likelihood gradient in the high-likelihood area and adopt the form of

$$\text{bonus} = \boldsymbol{y} \cdot \log(1 - \boldsymbol{p}), \tag{8}$$

which ensures that the monotonicity of the likelihood gradient is consistent with that of the Cross-Entropy Loss, meaning that the classifier obtains more bonus when making highly-confident correct predictions. The design is against most existing studies in that the high-likelihood area is given more focus than the low-likelihood area.

**Encouragement** implies the system realizes unusual achievements that should be encouraged. Since the unusual achievements in long-tailed classification should be correctly predicting rare class examples, we propose to reward the system based on the frequency of the example's class:

$$\text{encouragement} = w_y = \frac{\bar{m}}{m_y}, \tag{9}$$

where $m_y$ denotes the measurement of the frequency of the class $y$. The form is flexible and similar to the frequency-based methods, and thus can be further extended based on the related studies. In our experiments, we use the effective number from Cui et al. (2019) as the measurement.

Compared to the existing frequency-based and likelihood-based objective, our Eureka Loss takes the bonus term to calibrate the attention to different likelihood landscapes and the encouragement term to inform the model with the class difficulty, composing a more targeted yet comprehensive loss for learning imbalanced distributions.

## 5 EXPERIMENTS

We validate the proposed Eureka Loss on diverse long-tailed classification problems and analyze the characteristics of the Eureka Loss with insights into the learned models.

### 5.1 TASKS, DATASETS, AND TRAINING SETTINGS

**Tasks and Datasets** We conduct experiments on two image classification tasks and a dialogue generation task. **iNaturalist 2018** is a real-world dataset which embodies a highly imbalanced class distribution of 8,142 classes. Apart from the test performance, we also report the validation performance grouped by the class frequency and categorize the examples into three groups: many (classes with more than 100 examples), medium (classes with 20 to 100 examples), and few (classes with fewer than 20 examples). **ImageNet-LT** (Liu et al., 2019) is an artificially constructed long-tailed classification dataset based on ILSVRC 2012 of 1000 classes. **ConvAI2** is a natural conversation dataset for evaluating dialogue system, where each word type can be treated as a class, i.e., 18,848 words (classes) in total, and have extremely imbalanced training and test datasets.

**Evaluation Metric** For the image classification tasks, we use the **accuracy** on 'All' data and subset of classes with 'Many', 'Medium' and 'Few' examples , i.e., the precision of the top-1 prediction. Since the test set of those tasks are balanced in classes, we further propose to estimate the accuracy on the imbalanced class distribution that reflects natural performance in real-world scenarios. The **natural accuracy** is the linear interpolation of the accuracy on the balanced test set using the class frequencies from the training set. For the natural language generation task, we adopt the **micro and macro F-scores** from Zhang et al. (2018) between the generated and the reference sentences to check how well the systems participate in the conversation. We further adopt the **4-gram diversity** to examine the rare phrases, since a well-known problem for dialogue tasks is that the model tends to generate common, dull and repetitive responses and thus cannot capture the diversity of natural

| Method | All | Many | Medium | Few | All (Natural) |
|---|---|---|---|---|---|
| CE | 64.3 | **74.1** | 65.9 | 59.8 | 71.7 |
| CB | 58.3 | 61.8 | 58.6 | 56.9 | 59.8 |
| FL | 62.9 | 72.9 | 64.1 | 58.8 | 70.7 |
| FL + CB | 59.6 | 45.5 | 61.8 | 60.5 | 46.7 |
| Eureka Loss | 68.5 | 70.8 | 69.3 | 66.7 | 70.6 |
| CB[†] | 68.1 | 71.0 | 68.3 | 67.1 | 70.4 |
| FL + CB[†] | 66.9 | 64.4 | 67.4 | 67.0 | 64.9 |
| LDAM+CB[†] (Cao et al., 2019) | 63.3 | 65.2 | 63.0 | 63.1 | 64.1 |
| BBN (Zhou et al., 2019)[*†] | 69.6 | - | - | - | - |
| Decoupling-LWS (Kang et al., 2020)[*†] | 69.5 | 71.0 | 68.8 | 69.5 | - |
| Eureka Loss[†] | 69.9 | 73.3 | 69.3 | 69.6 | **73.0** |
| FSA (Chu et al., 2020)[*§] | 65.9 | - | - | - | - |
| MBJ (Liu et al., 2020)[*§] | 68.6 | 68.0 | 69.8 | 68.5 | - |
| Eureka Loss + CB[†] | **70.3** | 69.0 | **70.1** | **70.9** | 69.4 |

Table 2: Results on iNaturalist 2018. [*] denotes results from the corresponding paper and [†] denotes deferred learning, where the base loss is applied at the beginning of training and the improved method is adopted later. [§] denotes that it focuses on transferring representations and the method is less related to our work. Best results are shown in bold. The proposed Eureka Loss achieves the best results in learning both common and rare classes.

language distributions. Since the test set of natural language generation task is naturally imbalanced, we do not need to estimate the natural performance.

For the detailed introduction to tasks, datasets, and training settings, please refer to the appendix.

## 5.2 RESULTS

**iNaturalist 2018** The results are reported in Table 2. We tune the hyper-parameters for our implemented baselines and report the averaged performance among 3 runs at the best setting.

We compare Eureka Loss with frequency-based method Class-balanced Loss (CB), likelihood-based method Focal Loss (FL) and their combination FL+CB. As we can see from the first group in the table, neither FL nor CB achieves improvements over Cross Entropy (CE), but Eureka Loss outperforms CE by a large margin in terms of overall accuracy and accuracy for classes with few examples.

In contrary to the first group, considering only the accuracy on the balanced test set, the two-stage version of frequency class-balanced training which adopts the class-balanced training only in the latter training phase includes deferred CB(denotes CB[†] in the table), LDAM + deferred CB, BBN and Decoupling-LWS enjoy clear advantage over CE. In order to check whether Eureka Loss is additive with the deferred method and the class-balanced training, we take deferred Eureka Loss and Eureka Loss + CB[†] into comparison.

The deferred Eureka Loss is motivated by an intuition that when training enters the bottleneck stage, Eureka Loss is introduced to reward rare classes to encourage the model to learn less common patterns, which may be helpful for learning. Compared with the original method, the deferred encouragement brings improvement on both balanced and imbalanced test distribution (+1.4 and +2.4 regarding All and All(Natural), respectively). Moreover, the class-balanced training still impairs the learning for common classes even under the deferred setting, which may cast into unfavorable natural performance in real applications, e.g., the accuracy on the 'Many' subset for CB[†] and Decoupling-LWS is under-performs CE by 3.1, the results is that applying CB[†] reduces the Natural accuracy by 1.3. But deferred Eureka Loss largely outperforms CE and these methods on both balanced and imbalanced test distributions. The reason may be that we do not impair the CE learning and the additional rewarding for rare classes is less harmful. Since Eureka Loss only introduces an additive term, it is flexible and can be combined with CB, the adoption results in best All accuracy of 70.3.

| Method | All | All (Natural) |
|--------|-----|---------------|
| CE | 44.6 | 63.7 |
| FL | 43.6 | 62.4 |
| CB | 43.9 | 58.8 |
| FL+CB | 31.1 | 25.0 |
| Eureka Loss | **48.4** | **63.8** |

| Method (Deferred) | All | All (Natural) |
|-------------------|-----|---------------|
| OLTR (Liu et al., 2019)[*] | 46.5 | - |
| CB[†] | 49.2 | 60.5 |
| Decoupling-LWS (Kang et al., 2020)[*] | 49.9 | 59.8 |
| Eureka Loss + CB[†] | **50.4** | **62.2** |

Table 3: Results on ImageNet-LT. [*] and [†] are defined similarly to Table 2. Eureka Loss demonstrates consistent improvements against existing methods.

| Name | F-Score (Macro) | F-Score (Micro) | Diversity (4-gram) |
|------|-----------------|-----------------|--------------------|
| Cross Entropy | 1.17 | 16.9 | 36.5 |
| Focal Loss ($\gamma = 1$) | 1.17 | 16.8 | 36.6 |
| Focal Loss ($\gamma = 2$) | 1.13 | 16.7 | 37.3 |
| Eureka Loss | **1.32** | **17.2** | **40.4** |

Table 4: F-scores and 4-gram diversity on ConvAI2. The proposed Eureka Loss achieves better performance than baseline methods and generates responses that are more diverse.

In all, adopting the Eureka Loss achieves a balanced performance on both common and rare classes. Besides, we also outperform the latest representation transferring based methods including MBJ and FSA.

**ImageNet-LT**    Table 3 demonstrates the results on ImageNet-LT of various methods. For this artificial dataset, we first compare with the representative frequency-based method CB and likelihood-based method Focal Loss (FL). As we can see, the proposed method obtains a significant improvement in the balanced test set and also maintains the lead position in the virtual natural test set. For comparison with methods that defer the class-balanced training including deferred CB and Decouping-LWS, the Eureka Loss of corresponding modification also enjoys a comfortable margin and arguably excels in balancing the performance on both of the common and the rare classes.

**ConvAI2**    Table 4 shows that the proposal helps the prediction of rare words (+10% macro F-score) and thus improves the diversity of language generation (+10% 4-gram diversity). Since this dataset is extremely imbalanced, e.g., the imbalance ratio is over 200,000, the frequency-based methods require extensive tuning to work, which we thus omit from the comparison as we are not able to reproduce favorable results. Compared with the likelihood-based method Focal Loss, which is marginally better that the original cross-entropy loss, the Eureka Loss still obtains substantial improvements.

### 5.3    ANALYSIS

**Effect on Distributions of Different Imbalance Degrees**    To analyze the effect on imbalanced distributions of different degrees, we construct several artificial datasets based on CIFAR-100 and control the size of the rarest class. The imbalance degree stands for the ratio of the class size of the most common class to that of the rarest class. Hence, the larger the degree, the more imbalanced the dataset. For example, if imbalance degree is 100 for CIFAR-100, the most common class has 500 examples and the rarest class has 5 examples. As shown in Table 5, the Eureka Loss is consistently better than existing methods, especially for datasets that are more imbalanced. For results on CIFAR-10, please refer to the appendix.

| Method | ID-100 | ID-50 | ID-10 |
|--------|--------|-------|-------|
| CE+RS[†♣] | 41.61 | 46.48 | 58.11 |
| CE+CB[†♣] | 41.51 | 45.29 | 58.12 |
| LDAM+CB[†♠] | 42.04 | 46.62 | 58.71 |
| BBN[†♣] | 42.56 | 47.02 | 59.12 |
| EL + CB[†] | **43.19** | **48.48** | **59.31** |

Table 5: Results on long-tailed CIFAR-100 of different imbalance degrees (ID). [†] denotes deferred learning; [♣] and [♠] denotes results taken from Zhou et al. (2019) and Cao et al. (2019), respectively.

**Varying Strength of Bonus**    To illustrate the importance of the high-likelihood area in imbalanced classification within Eureka Loss, we compare a

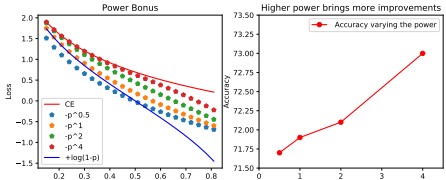

Figure 3: Varying strength of bonus on long-tailed CIFAR-10. Higher power $\gamma_b$ indicates higher strength.

| Method | All | Many | Medium | Few |
|---|---|---|---|---|
| EL ($\beta$=0.99) | 45.8 | **67.2** | 38.4 | 11.3 |
| EL ($\beta$=0.999) | 47.8 | 66.1 | 42.0 | 16.4 |
| EL ($\beta$=0.9999) | **50.0** | 67.1 | **44.7** | **19.8** |

Table 6: Varying strength of encouragement on the dev set of ImageNet-LT. Higher $\beta$ means higher strength.

exponential form bonus called **Power Bonus (PB)** to the original bonus, which takes the power form of the probability vector of by a factor $\gamma_b$:

$$PB(\boldsymbol{p}) = -\boldsymbol{y} \cdot \boldsymbol{p}^{\gamma_b}, \tag{10}$$

where $\gamma_b$ is a positive value to ensure the monotonicity and can be tuned for different tasks. Besides, CE achieves a 71.4% accuracy and the likelihood bonus with deferred encouragement gets a 76.1% accuracy. Figure 3 demonstrates that bigger likelihood gradient in the high-likelihood area brings more improvements, e.g., power-bonus with power of 4 is better than bonuses of small power.

**Varying Strength of Encouragement**  The strength of encouragement is determined by both of the class frequency and the hyper-parameter $\beta$ as we use the effective number of the class. As $\beta$ controls the variance of the the effective number, e.g., when $\beta = 0$, the variance is 0, meaning all of the classes receive equal encouragement, we control the strength of the encouragement towards tail classes by altering $\beta$. The results on the validation set of ImageNet-LT are shown in Table 6. As we can see, higher $\beta$ (more encouragement for tail classes), is connected to higher overall accuracy and better tail class performance, which again validates our motivation for encouraging correct rare class predictions.

**Effect on Example-Likelihood**  The Eureka Loss rewards the high-likelihood predictions especially for tail classes. It is interesting to see how the training dynamic is changed due to this preference. In order to understand the effect, we visualize the example likelihood grouped by target class frequencies after training in Figure 4 and Figure 5 (due to space limit, the complete comparison is provided in the Appendix A), which are from the validation set from iNaturalist 2018. As we can see, with the Eureka Loss, the examples in the high-likelihood area are driven to the extreme. For example, considering the medium and the low frequency group, the "hard" examples that may be inherently difficult to classify stand invariant, while for the examples that can be classified correctly, the system now treats them with more confidence. This dynamics translate into better accuracy in unseen examples in the test set, hinting the importance of rare class examples in the high-likelihood area for the generalization of learning imbalanced class distributions.

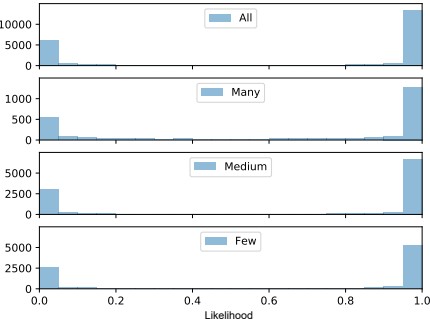

Figure 4: The visualization of test likelihood distribution for models trained with the Eureka Loss. The model classifies the rare class examples more decisively, compared to the existing methods.

## 6  CONCLUSIONS

In this paper, we examine the effect of the high-likelihood area on learning imbalanced class distributions. We find that the existing practice of relatively diminishing the contribution of the examples in the high-likelihood area is actually harmful to the learning. We further show that the rare class examples in the high-likelihood area have pivotal contribution to model performance and should be focused on instead of being neglected. Motivated by this, we propose the Eureka Loss, which additionally rewards the well-classified rare class examples. The results of the Eureka Loss in the image classification and natural language generation problems demonstrate the potential of reconsidering the role of the high-likelihood area. In-depth analysis also verifies the effectiveness of the investigated loss form and reveals the learning dynamics of different approaches to long-tailed classification.

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

## A   VISUALIZATION OF EXAMPLE LIKELIHOOD

The complete comparison between the likelihood distribution for Eureka Loss, Cross-Entropy Loss, Focal Loss, Class-balanced Loss are shown in Figure 5. We see from the figure that the model trained with Eureka Loss gives high-confidence predictions for gold labels. Compared with the Cross-Entropy Loss, the Focal Loss diminish the contribution of high-likelihood examples, and the resulted model is unsure in the prediction of unseen examples. In particular, for the examples in the Few group, it almost produces no confident correct predictions. The Class-Balanced Loss, on the other hand, improves the confidence for the tail class examples but degrade the performance for the head class examples, which may imply potential issues regrading to natural performance in real-world applications. Besides, it is worth noting that the Decoupling-LWS obtains a likelihood distribution similar to the Class-Balanced Loss.

## B   FURTHER RESULTS AND ANALYSIS

### B.1   RESULTS ON INATURALIST 2018 WITH TRAINING FOR 90 EPOCHS

It is found by existing work (Kang et al., 2020) that training much longer for the iNaturalist 2018 dataset can produce better scores and reflect the performance of the models more authentically. However, most previous studies conduct training for a shorter time. To keep consistent with previous research in this field, we also train the models using Eureka Loss for 90 epochs and the results are shown in Table 7. In this setting, eureka loss achieves better accuracy than the two-stage decoupling methods (Decoupling-LWS and BBN), the advantage is more profound under the Natural accuracy, for example, compared to the Decoupling-LWS, the deferred Eureka Loss gains 3.8 Natural accuracy. Compared to the one-state methods including Class-Balanced Loss (CB), Focal Loss (FL), Class-Balanced Focal Loss (FL+CB), LDAM, the model trained with the Eureka Loss is much more accurate on the test distribution.

### B.2   RESULTS ON LONG-TAILED CIFAR-10

In the main text, we have reported the results of the Eureka Loss varying the class imbalance on the CIFAR-100 dataset. Here we also perform comprehensive experiments on long-tailed CIFAR-10 and report top-1 precision on the balanced test set. The results are shown in the Table 8. When combined with Class-Balanced Loss, Eureka Loss brings higher improvement in terms of accuracy than Cross-Entropy Loss and LDAM.

### B.3   HYPER-PARAMETER OF THE FOCAL LOSS

In the paper, we report results for the Focal Loss with best hyper-parameters. For COCO detection, the hyper-parameter of $\alpha = 0.25, \gamma = 2$ is the best setting reported in Table 1.b of the origin paper

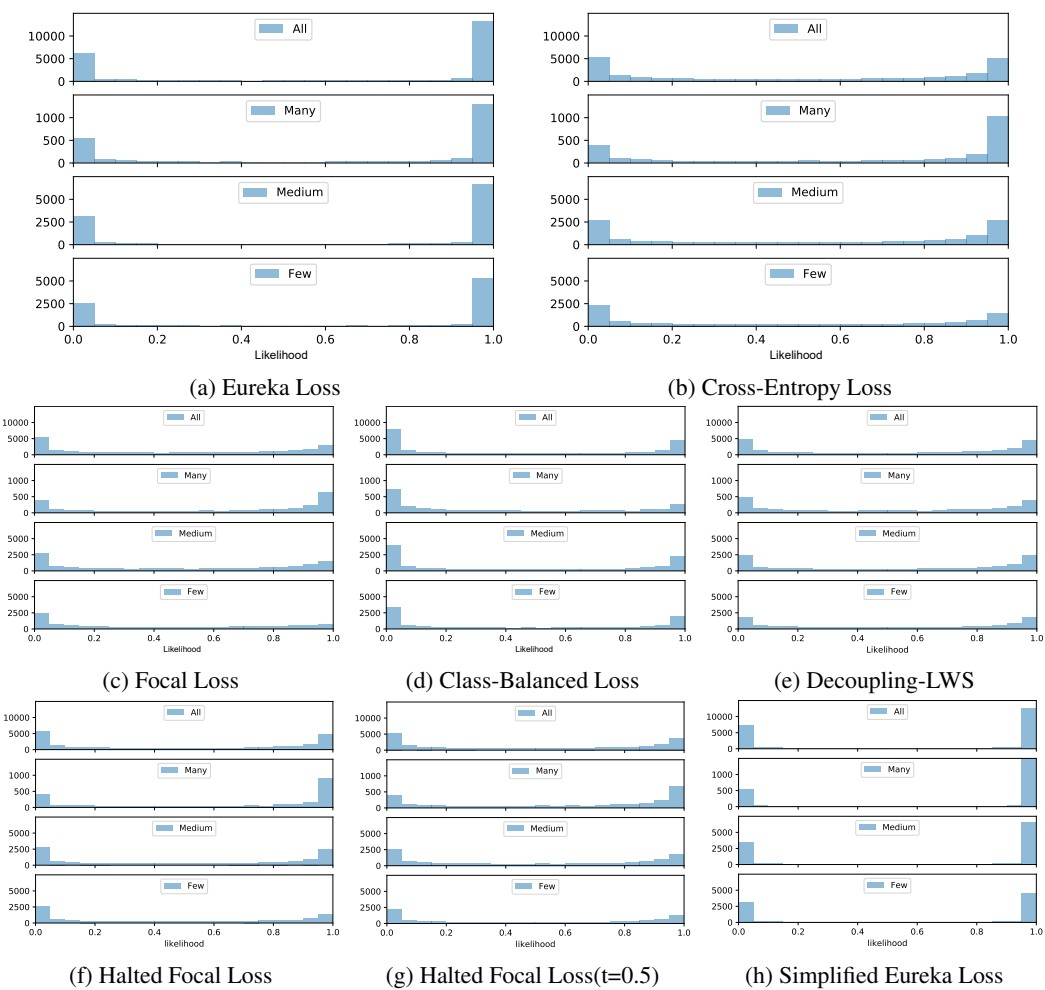

Figure 5: Example likelihood on the validation set of iNaturalist18 categorized by class frequencies into few, medium, many, and all.

(Lin et al., 2017). For the other multi-class classification tasks, we tune hyper-parameters of the Focal Loss on it. The accuracy for the Focal Loss with different hyper-parameter $\gamma$ are listed in the Table[1]. we set $\gamma = 1$ for Focal Loss since it is consistently optimal in long-tailed image classification. For ConvAI2, $\gamma = 0.5$ under-performs Cross Entropy, and neither $\gamma = 1$ nor $\gamma = 2$ outperforms each other, so we report the Focal Loss of $\gamma = 1$ and the Focal Loss of $\gamma = 2$ in the Table 4.

### B.4 COMPLEMENTARY EXPERIMENT TO THE MOTIVATION EXPERIMENT

In Section 3, we propose Halted Focal Loss(HFL) and compare it to Focal Loss(FL) to illustrate the potential of the high-likelihood area. However, its loss is no steeper than Cross Entropy (CE). Moreover, Focal Loss does not beat CE in the setting of multi-class classification. In order to bridge the gap between the possibly weak motivation experiment of Halted Focal Loss and the proposed method Eureka Loss. We propose simplified Eureka Loss:

$$\mathcal{L} = \begin{cases} -\boldsymbol{y} \cdot \log \boldsymbol{p}, & \text{if } p_y \leq \varphi \\ -\boldsymbol{y} \cdot \log \boldsymbol{p} + \boldsymbol{y} \cdot [\log(1 - \boldsymbol{p}) - b], & \text{otherwise} \end{cases} \tag{11}$$

---

[1]Following (Cui et al., 2019), we omit the hyper-parameter $\alpha$, since the Focal Loss with uneven 'alpha' for each class in the setting of multi-class classification can be viewed as Class-balanced Focal Loss (FL+CB) and FL+CB is compared individually.

| Method | All | Many | Medium | Few | All (Natural) |
|---|---|---|---|---|---|
| CE | 61.8 | **72.2** | 63.4 | 57.2 | **69.8** |
| FL | 60.2 | 70.4 | 61.5 | 56.0 | 68.3 |
| CB | 52.4 | 51.9 | 53.0 | 51.8 | 51.2 |
| RS | 56.7 | 59.0 | 56.9 | 55.9 | 56.5 |
| FL+CB (Cui et al., 2019)[*] | 61.1 | - | - | - | - |
| CE+CB[†] (Cao et al., 2019)[*] | 63.7 | - | - | - | - |
| LDAM (Cao et al., 2019)[*] | 64.6 | - | - | - | - |
| LDAM+CB[†] (Cao et al., 2019)[*] | **68.0** | - | - | - | - |
| LDAM+CB[†] (Zhou et al., 2019)[*] | 64.6 | - | - | - | |
| BBN (Zhou et al., 2019)[*] | 66.3 | - | - | - | - |
| Decoupling-LWS (Kang et al., 2020)[*] | 65.9 | 65.0 | 66.3 | 65.5 | 65.7 |
| Eureka Loss | 66.4 | 67.5 | 66.5 | 65.9 | 67.9 |
| Eureka Loss[†] | 67.1 | 69.4 | **67.3** | **66.1** | 69.5 |

Table 7: Results on iNaturalist 2018 after training the model for 90 epochs. [*] denotes results from the corresponding paper and [†] denotes that we use the base loss at the beginning of training and then adopt the method later. The proposed Eureka Loss achieves good results in learning both common and rare classes.

| Method | ID-100 | ID-50 | ID-10 |
|---|---|---|---|
| CE+RS[†♣] | 75.61 | 79.81 | 87.38 |
| CE+CB[†♣] | 76.34 | 79.97 | 87.56 |
| LDAM+CB[†♠] | 77.03 | 81.03 | 88.16 |
| BBN[†♣] | **79.82** | **82.18** | 88.32 |
| EL + CB[†] | 77.95 | 82.00 | **88.35** |

Table 8: Results on long-tailed CIFAR-10 data with different imbalance degrees. ID is short for imbalance degree. [†] is defined similarly; ♣ denotes Zhou et al. (2019); ♠ denotes Cao et al. (2019).

where $\varphi$ is set to 0.5, and $b$ is $log(1 - \varphi)$. In the simplified Eureka Loss, the encouragement is removed, and to keep the low-likelihood area unchanged, the new bonus term starts rewarding the model from $p = \varphi$. As is shown in Table 10, HFL(t=0.5) is also better than HFL and FL in terms accuracy of tail classes on the large scale long-tailed classification dataset iNaturalist 2018, This result once again shows that high-likelihood area matters and near-correct predictions of rare classes play a major role. But HFL is the combination of Focal Loss(in the low likelihood area) and Cross Entropy(in the high-likelihood area) and the performance is constrained. Unlike HFL, the loss of Simplified Eureka Loss is built on CE and the loss is much steeper than Cross Entropy in the high-likelihood area, it outperforms Cross Entropy(CE) and HFL in terms of all metrics, especially on the subset of tail classes. Eureka Loss reported in Table 2 is a continuous version of simplified Eureka Loss with an additional encouragement for rare classes, similar to HFL(t=0.5), this setting which rewards more for rare classes achieves the best overall performance.

## B.5 EUREKA LOSS MITIGATES OVER-FITTING ON TAIL CLASSES

As shown in Figure 6, compared to Cross-Entropy Loss, Eureka Loss reduce the gap between the training accuracy and test accuracy from 33.0 to 28.6 on tail classes. Moreover, even though Class-balanced Loss achieves the highest training accuracy, its test accuracy is unexpectedly low. The difference of performance between the seen examples and the unseen examples indicates the degree of over-fitting. The results are from the "Few" subset of the iNaturalist 2018 of training 90 epochs.

| $\gamma$ | CIFAR10 | Imagenet-LT | iNaturalist2018[2] |
|---|---|---|---|
| 0.5 | 21.2/10.8 | 8.3/7.1 | 9.2/3.2 |
| 1 | 70.8/0.6 | 43.8/0.3 | 60.2/0.3 |
| 2 | 70.3/0.6 | 43.6/7.9 | 59.5/0.2 |

Table 9: Mean/(standard deviation) test accuracy of the Focal Loss with different hyper-parameter $\gamma$ in long-tailed image classification.

| method | CIFAR10 Mean/Stdev | All | Many | iNaturalList2018 Medium | Few | All(natural) |
|---|---|---|---|---|---|---|
| CE | 71.4/0.5 | 64.3 | 74.1 | 65.9 | 59.8 | 71.7 |
| FL | 71.3/0.6 | 62.9 | 72.9 | 64.1 | 58.8 | 70.7 |
| HFL | 71.4/0.3 | 63.6 | 73.1 | 64.6 | 59.7 | 70.7 |
| HFL(t=0.5) | **71.8**/0.5 | 64.2 | 73.3 | 64.3 | 61.6 | 70.8 |
| Simplified EL | **71.8**/0.9(+0.4) | **66.3**(+2.0) | **75.1** (+1.0) | **67.4**(+1.6) | **62.6**(+2.8) | **72.6**(+1.1) |

Table 10: Comparison between the Halted Focal Loss and the Simplified Eureka Loss on long-tailed Cifar-10 (imbalance ratio is 100) and iNaturalist 2018, the mean standard deviation (Stdev) for results on iNaturalList2018 is about 0.4.

# C  DETAILS OF EXPERIMENTAL SETTINGS

## C.1  DATASETS

There are six datasets used in this paper in total and an overview of the dataset statistics are demontrated in Table 11 and Figure 7. For image classification tasks, the iNaturalist 2018 dataset is the most imbalanced and has the most classes, which is most satisfactory for evaluating long-tailed classifications. For the language generation task, ConvAI2 has an imbalance ratio of 277K, which, however, should be taken cautiously, since most of the tail classes are not covered in evaluation. The common practice to evaluate the learning on the imbalanced language distributions is to investigate the diversity of the generated text. The 4-grams can be regarded as high-ordered classes and a 4-gram of four common words can also be a "rare class".

## C.2  TRAINING SETTINGS

**CIFAR-10 and CIFAR-100**  For experiments on long-tailed CIFAR-10 and CIFAR-100, the backbone network is ResNet-32 (He et al., 2016). The model is optimized with SGD with a momentum of 0.9. The learning rate is set to 0.1 and the model is trained for 200 epochs with 128 examples per mini-batch. To stabilize the training, we adopt the warm-up strategy used by Goyal et al. (2017) in the first 5 epochs. Following Cao et al. (2019), we decay the learning rate by 0.01 at the 160th epoch and again at the 180th epoch. For the results in Figure 2 and Figure 3, we conduct experiments on long-tailed CIFAR-10 with an imbalance ratio of 10.

**ImageNet-LT**  For experiments on ImageNet-LT ILSVRC 2012, the base network is ResNext-50 (He et al., 2016). The batch size is set to 512 to accelerate training. The initial learning rate is 0.2 and we utilize a cosine learning rate scheduler.

**iNaturalist 2018**  Same as the experiments on ImageNet-LT, we also follow the default setting in Kang et al. (2020) for experiments on iNaturalist. To be specific, we adopt ResNet-50 model, use SGD with to train the model for 200 epochs with batch size 512 and a cosine learning rate schedule which gradually decays from 0.2 to 0.0. Results on the valid set are also reported on subset of many ($> 100$ samples), medium ($20 - 100$ samples), and few ($< 20$ samples), respectively.

**ConvAI2**  For the conversation generation task, we utilize a two-layer LSTM (Hochreiter & Schmidhuber, 1997) encoder-encoder architecture as our base network. The hidden size of both encoder and decoder is set to 1024. We optimize the model with SGD optimizer with momentum 0.9, the

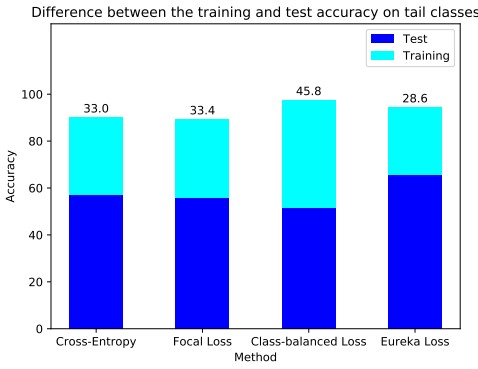

Figure 6: Illustration of the over-fitting phenomenon on tail classes, the number on the top of each bar is the difference between the training accuracy and the test accuracy.

| Dataset | # Classes | Imbalance Ratio |
|---|---|---|
| COCO Detection | 2 | ~1000 |
| Long-tailed CIFAR-10 | 10 | 10-100 |
| Long-tailed CIFAR-100 | 100 | 10-100 |
| ImageNet-LT | 1000 | 256 |
| iNaturalist 2018 | 8142 | 500 |

Table 11: Data statistics of long-tailed image classification tasks. Imbalance Ratio denotes the ratio of the size of the most common class to that of the rarest class.

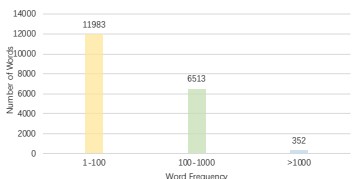

Figure 7: Word frequency distribution of ConvAI2 dataset. For natural language generation tasks, each word type can be regarded as a class and most words appear scarcely in the data. If measured the same with long-tailed image classification tasks, the imbalance ratio is 277K.

batch size is 64 and the learning rate is set to 3. The embedding size is 256 and word vectors are initialized with GloVe (Pennington et al., 2014). We select our final model until the performance on the validation is no longer improving after 5 epochs.

**COCO Dectection** For experiments on COCO detection, we adopt the configuration of "RetinaNet-R-50-FPN-1x" from the GitHub repository Detectron2 as our default setting. In this setting, the one-stage detector of RetinaNet with the backbone ResNet50 is trained for 90k updates and the batch size is 8 images per batch.

For the image classification tasks, the default $\beta$ is set to 0.9999 for all datasets. For the deferred version, we defer the adoption of Eureka Loss after training for 160 epochs and 180epochs on CIFAR100 and iNaturalist 2018 respectively. As for the dialog generation task ConvAI2, $\beta$ is set to 0.999 and we start the encouragement after regularly training the model for 5 epochs.

We tune the $\beta \in \{0.9, 0.99, 0.999, 0.9999\}$ and $\gamma \in \{0.5, 1, 2\}$ for the Class-Balanced Loss (CB) and the Focal Loss (FL) respectively in multi-class classification, and report the best results of these baselines. Following previous work (Cui et al., 2019), $\alpha$ is set to 1.0 for the Focal Loss(FL), and the Class-Balanced Focal Loss (FL+CB) in multi-class classification tasks can be viewed as the origin Focal Loss with class-level weight $\alpha$ in binary classification tasks.

The training costs are summarized in Table 12.

| Data | Infrastructure | Mem/GPU | Time | Epochs | Samples | Model |
|------|----------------|---------|------|--------|---------|-------|
| Long-tailed CIFAR-10 | RTX 2080Ti * 1 | 1.4G | 0.3h | 200 | 12.4K | ResNet32 |
| Long-tailed CIFAR-100 | RTX 2080Ti * 1 | 1.4G | 0.3h | 200 | 10.8K | ResNet32 |
| ImageNet-LT | TITAN RTX * 4 | 17G | 8h | 90 | 116K | ResNext50 |
| iNaturalist 2018 | RTX TITAN * 4 | 15G | 48h | 200 | 438K | ResNet50 |
| ConvAI2 | RTX 2080Ti * 1 | 9G | 4h | 16 | 131K | 2-L LSTM |
| COCO Detection | RTX 2080Ti * 4 | 9G | 8h | 90K updates | 118K | ResNet50 |

Table 12: Training costs of each task. Samples are dialogues in ConvAI2 and images in image classification tasks

