# OpenReview forum: "High-Likelihood Area Matters --- Rewarding Correct,Rare Predictions Under Imbalanced Distributions"
_ICLR.cc/2021/Conference — Reject_

### Official Review · AnonReviewer4 · 2020-10-23
**interesting finding but lacking explanation/understanding**

**Rating:** 5
**Confidence:** 5

**Review:**

Summary:

- This paper made a finding that weighting up  correct predictions for rare class examples also can help to improve the performance of imbalanced classification. In light of this finding, it proposes the Eureka Loss to add additional gradients for examples belong to rare classes in the high-likelihood area when correctly predicted. Experiments on several large-scale benchmarks demonstrate its effectiveness.

Pros:
- The paper is clearly written and easy to follow.
- The experiments are thorough and demonstrate the effectiveness.

Cons:
- While the finding is quite interesting, I think the design of the proposed algorithm is quite arbitrary. It's not clear to me why the authors choose to add a term for rare classes rather than changing the weights directly. Why don't the authors use HFL in the end?
- Currently it seems that there lacks complementary theory/intuition that could explain why weighting up the already correctly classified rare examples help with the performance.

Additional Questions:
- Figure 4 seems quite interesting. It seems that the functionality of Eureka Loss is quite different from HFL. I could intuitively understand that the Eureka loss function would encourage the examples to have likelihood of either 1 or 0. Have the authors visually checked the examples with a likelihood of 0? Does that mean training on a carefully selected subset gives better performance?

----
post-rebuttal update

I thank the authors for the responses. While I still think the idea is potentially interesting and original, I could not increase the score given the fact that this manuscript is naturally incremental without theoretical justifications.

---

> ### Author Response · Authors · 2020-11-25
> **Explain the idea and the transition from HFL to Eureka Loss**
>
> Explain the idea:
>
> As shown in Figure 5, the example of inaccurate penalty prediction is also rewarded with accurate prediction. The loss in the high likelihood area becomes steeper. This induces model gives prediction near 0 or 1, and the decision becomes clearer, which may increase the generalization ability of the model. The reward gives rare classes more encouragement, which makes the model get rid of the learning dilemma and be encouraged to learn more difficult patterns.
>
>
>
> Explain the  transition from HFL to Eureka Loss:
>
> Eureka Loss is steeper in high-likelihood area  than HFL, it rewards more than HFL for correct predictions and achieves better performance.   We  have added a subsection B.4 to expain the motivation for using EL.
>
> In section 2.2, we propose Halted Focal Loss(HFL) and compare it to Focal Loss(FL) to illustrate the potential of the high-likelihood area. However, its loss is no steeper than Cross Entropy (CE). Moreover, Focal Loss does not beat CE in the setting of multi-class classification. In order to bridge the gap between the possibly weak motivation experiment of Halted Focal Loss and the proposed method Eureka Loss. We perform the experiments of HFL on large-scale long tailed image classification task iNaturalist 2018 and propose simplified Eureka Loss in the rebuttal revision (subsection B.4 ‘Complementary experiment to the Motivation Experiment’)
>
> In the simplified Eureka Loss, the encouragement is removed, and to keep the low-likelihood area unchanged, a new bonus term starts rewarding the model from p=0.5, so it is a piece-wise function like HFL, but the loss in high-likelihood area is even steeper than CE. As is shown in Table 10, HFL(t=0.5) is also better than HFL and FL in terms accuracy of tail classes on the large scale long-tailed classification dataset iNaturalist 2018, This result once again shows that high-likelihood area matters and near-correct predictions of rare classes play a major role.
>
> But HFL is the combination of Focal Loss (in the low likelihood area) and Cross Entropy (in the high-likelihood area) and the performance is constrained. Unlike HFL, the loss of Simplified Eureka Loss is built on CE and the loss is much steeper than Cross Entropy in the high-likelihood area, it outperforms Cross Entropy(CE) and HFL in terms of all metrics, especially on the subset of tail classes. Eureka Loss reported in Table 2 is a continuous version of simplified Eureka Loss with an additional encouragement for rare classes, similar to HFL(t=0.5), this setting which rewards more for rare classes achieves the best overall performance. We hope the results of simplified Eureka Loss could explain the transition from HFL to Eureka Loss  in the paper.
>
>
>
> We have updated the Figure 5 the test likelihood distribution after training with each method, and we have included HFL in it.

---

### Official Review · AnonReviewer2 · 2020-10-28
**The experimental setting needs to be clarified**

**Rating:** 5
**Confidence:** 3

**Review:**

This paper deals with learning imbalanced class distributions.  First, it empirically finds that the high-likelihood area for the rare classes benefits classification. Then, based on the findings, it proposes a new learning objective called Eureka Loss, which can be viewed as a combination of the frequency-based and likelihood-based methods to reward the classifier when examples belong to rare classes in the high-likelihood area are correctly predicted. Empirical results on two typical tasks (i.e. image classification and language generation tasks) illustrate its superiority compared with other baselines.


###########################################################################################
pros:
1. Overall, it is well-written.
2. It clearly discusses the existing two methods (i.e. frequency-based methods and likelihood-based methods). Furthermore, it highlights the limitation of likelihood-based methods that they neglect the correctly-predicted tail class examples.
3. The motivation for the design of the new learning objective(i.e., Eureka Loss) is based on the empirical finding that the high-likelihood area of the rare examples is important to improve the performance.

###########################################################################################
cons:
1. The finding is mainly on empirical observations, which may lack theoretical support. Why is the high-likelihood area of the rare examples is important for generalization?
2. For the experimental settings, e.g. iNaturalist 2018, the i.i.d. assumption does not hold for the training and test set.
3. For the experimental results, how to tune the hyperparameter of the Eureka Loss, in validation set or test set? Since the reason in 2, I guess the hyper-parameter selection becomes difficult.

Minor comments:
For the last subfigure in Figure 1, the ordinate value for the loss is negative, which is wrong.

---

> ### Author Response · Authors · 2020-11-25
> **Explain the  idea behind  Eureka Loss and  clarify the issue about hyper-parameters.**
>
> Explain the idea: As shown in Figure 5, the example of inaccurate penalty prediction is also rewarded with accurate prediction. The loss in the high likelihood area becomes steeper. This induces model gives prediction near 0 or 1, and the decision becomes clearer, which may increase the generalization ability of the model. The reward gives rare classes more encouragement, which makes the model get rid of the learning dilemma and be encouraged to learn more difficult patterns.
>
> Hyperparameter : The selection of hyperparameter is much easier than related methods like CB, e.g. the beta is set to 0.9999 for all long-tailed image classifcations but  it should be tuned for every distribtion in their paper.
>
> Negative loss is not Wrong,  loss  can be negative and has been variously called a reward function, a profit function, a utility function, a fitness function in previous work.   Moreover, we can also add a constant to keep it postive, the sign does not matter but the monotonicity and the Steepness matter. We introduce the bonus item to encourage the model instead of penalizing the model for rare classes, and the bonus proves effective in our paper, either for Eureka Loss in Table 2  or Simplified Eureka Loss in Table 10.  Moreover, we can see from the Figure 4 and  Figure 5 that models trained with bonus for correct predictions make clearer decisions.

---

### Official Review · AnonReviewer3 · 2020-10-28
**Intriguing submission; experiments might be improved**

**Rating:** 5
**Confidence:** 3

**Review:**

The submission makes an intriguing claim that retaining focus on correctly predicted rare classes can improve performance for training with class-imbalanced datasets.

To illustrate this claim, the paper shows that one can find improvements at overall accuracy if a combination of the Focal Loss (which weights down examples with high predictive likelihood) and the cross-entropy loss is used such that the loss transitions from the Focal Loss to the CE for examples with predictive confidence above a threshold, for examples belonging to the top rarest classes. On long-tailed CIFAR-10, this produces a mild improvement at overall accuracy (around 0.5%) when the top 40% rarest classes receive this mixed loss. Further experiments with COCO-detection finds sparse improvements (around 0.4%) when applying the mixed loss to the tail classes.

Based on the above findings, the paper argues for not weighting down confident predictions, especially if these belong to rare classes. However, perhaps these experiments are insufficient to arrive at such a conclusion? To ensure that the minor improvements in CIFAR-10 are in fact due to the claimed reasoning, one could also look at other combinations of the losses that do not conform to the claim. For example, apply the loss to the top k% most confident examples (without stratifying by rare classes), randomly select k% of images, etc. For COCO-detection, apply HFL to the head classes, and FL to the tail classes. Since improvements are so small, it would also be nice to see some standard deviation bars over multiple trials. Also, were the choices of Focal Loss hyper-parameters made to elicit their best performance? From Figure 2, it looks like it underperforms the cross-entropy loss.

The paper proposes a new loss meant to "reward the well-classified rare examples”. This augments the cross-entropy loss with a log(1-p_y) term scaled with a number that reflects the frequency of class y, such that rarer classes are scaled higher.

Experiments have been conducted on 2 image classification datasets and 1 dialogue dataset. In all cases, the proposed loss appears to result in improvements over baselines.

Some questions/comments about the experiments:
 - It appears that the proposed loss performs particularly well when combined with CB. Are the competing methods also similarly augmented?
 - For Table 4, why is the Focal Loss only evaluated for 2 settings of gamma? Shouldn’t there be a hyper-parameter search and the best gamma used?
 - There are a lot of comparisons, with a lot of numbers being taken from past reported results. For all such comparisons, has it been ensured that the architectural and training details are fixed across comparisons? Otherwise the comparisons might not be fair, especially given that reported improvements are minor.
 - Especially when improvements are minor, it becomes important to look at aggregate numbers, so I’d suggest reporting standard deviations over multiple trials for all experiments.

Some typos:
“down applications” —> “downstream applications”
“a effective number” —> “an effective number”
“thus the likelihood” —> “so that the likelihood”
“deferred courage” —> “deferred encouragement"

Overall, the paper is clearly written and reports exhaustive experiments (with the caveats/questions above). While the motivating experiments in Section 2.2 are not very compelling, in part due to the very minor improvements, the key intuition that the classification of rare-class hard examples should be continued to be encouraged (so that their predictive confidence doesn’t drop as these examples are weighted down by some of the other methods) sounds interesting, although some of the phrasing about “rewarding well-classified examples” can be a bit awkward. My main concerns as of now are about experimental details, which are described above in the questions.

Post Rebuttal:
Thanks to the authors for responding. I'm still not sure if the experiments are particularly compelling. There appear to be differences amongst the baselines with regards to class balancing, and the motivating section is still weak; there are new experiments on a larger dataset, but now with a different loss (simplified EL) which is close enough to the proposed loss that this does not work very well as a motivation anymore. Apart from this, taking some of the comments from the other reviewers and the authors' responses into account, I am retaining my initial rating.

---

> ### Author Response · Authors · 2020-11-25
> **Part1: Response  to the questions about comparisons and  we have included a complemetary experiment with bigger improvements and large-scale dataset to stenghen the motivation.**
>
> 1.	Hyper-parameters of Focal Loss:
>
> Response:
> As described in the Appendix C.2--‘Training settings’, we have tuned the hyper-parameters of the Focal Loss, and we report its averaged accuracy with best hyper parameters.  For example, as we can see from the Tabel 9 in Appendix that ‘gamma=1’ is the optimal hyper-parameter for long-tailed image classification. For COCO detection task where the Focal Loss was proposed, we use the optimal setting of ‘alpha=0.25, gamma=2’ which achieves the best performance in Table 1.b of the origin paper after searching hyper-parameters (Lin et al. 2017). For Focal Loss on Convai2, since neither ‘gamma=2’ nor ‘gamma=1’ outperforms each other, we report both in Table 4. The hyper-parameters for Focal Loss is also discussed in Appendix B.3.
>
> 2.	Motivating experiments in Section 2.2 are not very compelling, in part due to the very minor improvements:
>
> Response:
>
> We reported mean performance of several runs in the Section2.2 using the optimal setting discussed in the response 1, in section 2.2, we propose Halted Focal Loss(HFL) and compare it to Focal Loss(FL) to illustrate the potential of the high-likelihood area. However, its loss is no steeper than Cross Entropy (CE). Moreover, Focal Loss does not beat CE in the setting of multi-class classification.
>
> In order to bridge the gap between the possibly weak motivation experiment of Halted Focal Loss and the proposed method Eureka Loss.  We perform the experiments of HFL on large-scale long tailed image classification task iNaturalist 2018 and propose simplified Eureka Loss in the rebuttal revision (subsection B.4 ‘Complementary experiment to the Motivation Experiment’)
> In the simplified Eureka Loss, the encouragement is removed, and to keep the low-likelihood area unchanged, a new bonus term starts rewarding the model from p=0.5, so it is a piece-wise function like HFL, but the loss in high-likelihood area is even steeper than CE.
>
> As is shown in Table 10, HFL(t=0.5) is also better than HFL and FL in terms accuracy of tail classes on the large scale long-tailed classification dataset iNaturalist 2018, This result once again shows that high-likelihood area matters and near-correct predictions of rare classes play a major role. But HFL is the combination of Focal Loss (in the low likelihood area) and Cross Entropy (in the high-likelihood area) and the performance is constrained. Unlike HFL, the loss of Simplified Eureka Loss is built on CE and the loss is much steeper than Cross Entropy in the high-likelihood area,  it outperforms Cross Entropy(CE) and HFL in terms of all metrics, especially on the subset of tail classes.
>
> Eureka Loss reported in Table 2 is a continuous version of simplified Eureka Loss  with  an additional encouragement for rare classes, similar to HFL(t=0.5), this setting which rewards more for rare classes achieves the best overall performance.
> We hope the results of HFL on iNaturalist 2018 and the results of simplified Eureka Loss make the motivation stronger

---

> ### Author Response · Authors · 2020-11-25
> **Part2:  Details about baselines and comparisons**
>
> Response:
>
> Due to the space limit, we did not explain baselines and comparisons in detail in the initial submission. Now we have included more details about baselines and comparisons in Section 2 and Section 5.2 of the revision.
>
> a) Are the baselines augmented with CB? BBN and Decoupling-LWS do not use In the old version, we described BBN (Zhou et al., CVPR20) and Decoupling-LWS (Kang et al., ICLR20) in the second paragraph in the Section ‘Related Work’, they are both state-of-the-art class-balanced methods (CB) and use CE at the beginning of the training, and they belong to deferred CB in general. Therefore, the comparisons between deferred EL, deferred CB+ EL and deferred CB are fair, and the augmented EL are better than these advanced class-balanced methods. We report the performance of deferred CB+EL to show that our method is additive with the deferred class-balanced training.
>
> The comparison between CB and EL as well as deferred CB and deferred EL show that rewarding correct predictions for tail classes not only less impair the learning of head classes but also learn better for tail classes compared to penalizing more for tail classes. MBJ and FSA are recently proposed state-of-the-art feature transferring method in this field and they are less related to our work, we take them into comparison to keep completeness of comparisons, in this comparison our method deferred EL and deferred CB+EL surpasses them by a large margin. By the way, MBJ is augmented with class re-balancing strategy and FSA is a two-stage method in which the standard CE training is adopted in the first phase.
>
> b) Architectural and training details are described in C 3.2 ‘Training settings’, we fix these variables in our experiments.
>
>  c) The standard deviation In this paper, we have run several experiments and reported the mean value. The average standard deviations for the main resutls on Cifar10, Cifar100, Coco Detection, ImageNet LT and iNaturalist 2018 are about 0.6, 0.6, 0.2, 0.3, 0.7, 0.3, they are relatively small. We accept your suggestion, and we have reported some of results including standard deviation in Table 9 and table 10 in the rebuttal revision. We will report the detailed deviation for each result in the final version. Besides, although the improvements of HFL on FL are relatively small, the improvements of Eureka Loss on Cross Entropy and other baselines are big.

---

### Official Review · AnonReviewer1 · 2020-10-29
**The motivation is not convincing enough.**

**Rating:** 4
**Confidence:** 3

**Review:**

These are several concerns:
1. In the view of motivation, I don't think the motivation is strong enough and is convincing. Also, I don't think rewarding correct predictions but not penalizing incorrect ones is a reasonable way. In my opinion,  rewarding the correct predictions may be a good way, but penalizing the incorrect ones should also be important.
2. In the view of experiments, though the authors add Table 7 in the appendix, which is the result for training 90 epochs, I still doubt why Eureka Loss does not work better than recent works when training 200 epochs (which is also a common setting recently). And it seems that using CE at the beginning of training is important, and +CB$^+$ works the best.  Moreover, In table 2, the results on "few" are especially not very good comparing with others, which makes it harder for me to believe that rewarding the high-likelihood area really matters a lot for tail classes. It seems that the experiment results are not strong enough to support the proposed opinion.

---

> ### Author Response · Authors · 2020-11-25
> **The first concern may originate from the misunderstanding and we have included more details about baselines and comparisons in the rebuttal revision to address the second concern.**
>
> 1.	As to our motivation, we do not claim that not to penalizing the incorrect predictions, our motivation is that the role of high-likelihood area is overlooked and we should increase their relative importance to the low-likelihood area by rewarding the correct predictions in the meanwhile. Both HFL and EL include this idea, and neither reduces the penalty for inaccurate predictions.
> As shown in the left subfigure in the Figure 2 and defined in the Formula 6, we compare the Halted Focal Loss (HFL) to the Focal Loss (FL) to demonstrate the relative importance of high-likelihood area, but we do not change the loss landscape in the low-likelihood area and thus do not stop penalizing incorrect predictions.  We plot Eureka Loss (EL) in the right subfigure of Figure 1 and its variants in the left subfigure of Figure 3, and define EL generally in the Formula 7, the additional bonus in the EL does not alleviate the penalization for incorrect predictions, and it only strengthens the relative importance of optimization in the high-likelihood area.
> 2.	Due to the space limit, we did not explain baselines and comparisons in detail in the initial submission. Now we have included more details about baselines and comparisons in Section 2 and Section 5.2 of the rebuttal revision.
> In the old version, we described BBN (Zhou et al., CVPR20) and Decoupling-LWS (Kang et al., ICLR20) in the second paragraph in the Section of Related Work, they are both state-of-the-art class-balanced methods (CB) and use CE at the beginning of the training, and they belong to deferred CB in general.
> Therefore, the comparisons between deferred EL, deferred CB+ EL and deferred CB are fair, and the augmented EL are better than these advanced class-balanced methods. We report the performance of deferred CB+EL to show that our method is additive with the deferred class-balanced training.
> The comparison between CB and EL as well as deferred CB and deferred EL show that rewarding correct predictions for tail classes not only less impair the learning of head classes but also learn better for tail classes compared to penalizing more for tail classes.
> MBJ and FSA are recently proposed state-of-the-art feature transferring method in this field, we take them into comparison to keep completeness of comparisons, in this comparison our method deferred EL and deferred CB+EL surpasses them by a large margin.

---

### Author Response · Authors · 2020-11-25
**We will release the code and a example of implementation for Eureka Loss is in the Supplemetary Material**

The pytorch version of Eureka Loss is available in the Supplemetary Material,  this new criterion is easy to plug in your code.

---

### Decision · Program_Chairs · 2021-01-07
**Final Decision**

**Decision:**

Reject

**Comment:**

This submission got 1 reject and 3 marginally below the threshold. The concerns in the original reviews include (1) lack of theoretical justification. The motivation and claim are from empirical observation; (2) the performance improvement is minor compared with the existing methods; (3) some experiment settings and details are not explained clearly. Though the authors provide some additional experiments to the questions about the experiments, reviewers still keep their ratings. The rebuttal did not address their questions. AC has read the paper and all the reviews/discussions. AC has the same recommendation as the reviewers. The major concerns are (1) the theoretical justification is not clear. The additional explanation given by the authors in their rebuttal, i.e., the prediction becomes sharper and thus the model generalization ability can be improved, is not justified. (2) the experiments are not very convincing and can be further improved in the following two aspects: (1) the motivation experiments should be conducted in a consistent manner, instead of using simplified EL in some cases; (2) the effectiveness of EL should be more significant otherwise it is not clear whether the claim is true or not. At the current status of this submission, AC cannot recommend acceptance for the submission.